# Two *Avastrovirus* Species Discovered in Psittaciformes Expand the Host Range of the Family *Astroviridae*

**DOI:** 10.3390/v17030450

**Published:** 2025-03-20

**Authors:** K9 Jenns, John-Sebastian Eden, Annabelle Olsson, David Phalen

**Affiliations:** 1Centre for Virus Research, Westmead Institute for Medical Research, Westmead, NSW 2145, Australia; js.eden@sydney.edu.au; 2Sydney Institute for Infectious Diseases, School of Medical Sciences, The University of Sydney, Sydney, NSW 2050, Australia; 3Sydney School of Veterinary Science, The University of Sydney, Camden, NSW 2570, Australia; annabelle.olsson@sydney.edu.au (A.O.); david.phalen@sydney.edu.au (D.P.)

**Keywords:** astrovirus, virus discovery, metatranscriptomics, parrots, Psittaciformes

## Abstract

Metatranscriptomics has recently revealed greater species richness and host range of the *Avastrovirus* genus, quadrupling the number of avian orders known to host them in less than a decade. Despite this growing awareness of astrovirus presence in wild birds, limited attention has been paid to these viruses in the context of disease in Australian avifauna. Here we used unbiased RNA sequencing of intestinal samples from a galah (*Eolophus roseicapilla*) and an Australian king parrot (*Alisterus scapularis*) with a chronic diarrhoeal and wasting disease to detect the entire genomes of two novel astrovirus species. We propose naming these viruses *Avastrovirus eolorosei* (PQ893528) and *Avastrovirus aliscap* (PQ893527). The phylogenetic positions of these viruses highlight the importance of current and future metatranscriptomic virus screening in investigations of avian host landscapes beyond *Galloanserae*. This is also the first documentation of avastrovirus infections in Psittaciformes and the first to report their potential role as disease agents in them.

## 1. Introduction

Astroviruses are positive sense nonenveloped RNA viruses. They have 6.1–7.3 kb long genomes with three open-reading frames that encode a serine protease and accessory non-structural proteins (ORF1a), RNA-dependent RNA polymerase (RdRp, ORF1b), and capsid precursor protein (CaP, ORF2). There are currently two genera within the *Astroviridae* family: *Mamastrovirus* from mammalian hosts and *Avastrovirus* from avian hosts, with increasing evidence for a clade within or near this genus of viruses that infect frogs and reptiles [1,2]. The first described avastroviruses were predominately associated with poultry species and their wild counterparts [3]. It is now known that avastroviruses also infect members of the families Charadriiformes [4,5], Columbiformes [6,7], and Passeriformes [8,9]. Additionally, individual cases of novel avastroviruses have also been identified in an Adélie penguin (*Pygoscelis adeliae*, Sphenisciformes) [10], a broad-billed prion (*Pachyptila vittata*, Procellariiformes) [11], and a red-crowned crane (*Grus japonensis*, Gruiformes) [9]. Since these discoveries, it has been suggested that there are likely many novel avastroviruses circulating in wild and urban avifauna that are yet to be described [5,12].

In intensively raised poultry, avastroviruses cause enteritis in turkeys, nephritis in chickens, and hepatitis in ducks [13,14,15,16,17]. In contrast, most avastroviruses that were detected in wild birds were detected in birds that were apparently healthy or had died from unknown causes, the exception being an astrovirus detected in brain tissues from several species of Australian passerines exhibiting neurological signs [12].

In this report, we describe the complete sequences of two novel avastroviruses, one from a wild recently fledged galah (*Eolophus roseicapilla*) and a second from a wild recently fledged Australian king parrot (*Alisterus scapularis*). Both parrots were suffering from a previously described syndrome characterized by chronic diarrheal disease resulting in severe weight loss and ultimately death. The syndrome occurs in galahs, Australian king parrots, long-billed corellas (*Cacatua tenuirostris*), and sulfur-crested cockatoos (*Cacatua galerita*) [18,19,20,21]. The cause of this disease may be multifactorial, with birds being typically concurrently infected with one or more of the following: the immunosuppressive psittacine beak and feather disease virus, a yeast (*Macrorhabudus ornithogaster*), a single-celled parasite (*Spironucleus* species), and various bacterial pathogens. Additionally, on the two occasions where electron microscopy has been used to examine the droppings of affected birds, one or more viruses with a morphology consistent with that of an astrovirus have always been identified [18,19].

## 2. Materials and Methods

### 2.1. Case Histories and Necropsy

Five recently fledged sub adult galahs (*Eolophus roseicapilla*) were submitted to the Avian Reptile and Exotic Pet Hospital of the University of Sydney (Tharawal country*) in January 2021. All were rescued from the same location in the Macarthur region of NSW, Australia. All individuals had similar signs, including severe pectoral muscle atrophy and fetid diarrhoea that was liquid and contained gas. All birds were euthanized following induction with isoflurane gas (Dechra Veterinary Products) in oxygen and then given pentobarbitone (Lethabarb) diluted 1:4 with saline slowly IV to effect. A complete necropsy was performed on one individual and the entire gastrointestinal tract was aseptically resected and stored at −80 °C until further processing.

The recently fledged Australian king parrot included in this study was 1 of 40 Australian king parrots (*Alisterus scapularis*) that presented to the Boongarry Veterinary Services, Cairns (Gimuy, Yidinji country*) during the summer of 2021. The bird was exhibiting signs of weight loss and diarrhoea identical to those seen in the galahs. It was euthanised because of its poor prognosis. The small and large intestines were aseptically removed and stored at −20 °C until further processing.

The material used in this study was submitted for diagnostic purposes. The Animal Ethics Committee at the University of Sydney was informed that findings from the diagnostic material were to be used in a publication and a waiver of ethics approval granted.

### 2.2. Tissue Processing, RNA Extraction, and Library Preparation

Samples from each case were handled, processed, and analysed separately. Tissues were defrosted on dry ice and a 5 mm^3^ cross section of the small intestine was placed in 2 mL of DNA/RNA Shield (Zymo Research, Irvine, CA, USA). These were homogenised for two minutes using a TissueRuptor (QIAGEN) then centrifuged at 8000 RCF. Clarified homogenates were taken forward for total nucleic acid extraction with ZymoBIOMICS DNA/RNA Miniprep Kit. We prepared RNA libraries for high throughput sequencing using an established workflow–RAPID*prep* [22]. Briefly, samples underwent ezDNase treatment (Invitrogen), FastSelect HMR & Bacteria & globin mRNA removal (QIAGEN), single-strand cDNA synthesis with SuperScript^TM^ IV VILO^TM^ Master Mix (Invitrogen, Waltham, MA, USA), and second-strand cDNA synthesis with Sequenase (Thermo Fisher Scientific, Waltham, MA, USA). ds-cDNA was cleaned up with Mag-Bind^®^ Total Pure NGS (Omega Bio-tek, Norcross, GA, USA). Library preparation included tagmentation and indexation using Nextera XT DNA Library Prep Kit with unique dual indexes (Illumina, San Diego, CA, USA).

### 2.3. High Throughput Sequencing and Metatranscriptomic Analysis

To ensure accurate pooling, we sequenced libraries together on a single iSeq (Illumina) run and used percent passing filter as a quality check to calculate appropriate input volumes for downstream sequencing. The resulting library pool was sequenced on a NovaSeq X platform (Illumina) by the Australian Genomic Research Facility (Melbourne/Naarm, Wurundjeri country*) using a 10B (300 cycle) flow cell. Each library was sequenced targeting a minimum read depth of 20 million paired reads. Raw reads were processed through an in-house metatranscriptomic analysis pipeline. Briefly, we performed host removal of the galah library using an index of the galah reference genome (GCA_013397615) Bowtie2 v2.4.4 [23] and samtools v1.9 [24]. There is currently no genome available for the Australian king parrot, so we built a Bowtie2 index from the reference genome of the Alexandrine parakeet (*Psittacula eupatria*, GCA_031762335) and applied the same approach. Host-filtered reads were trimmed and deduplicated using fastp v0.22.0 [25] and quality filtered to remove low-complexity sequences with prinseq v0.20.4 [26]. We then sorted and selected non-ribosomal reads using sortmeRNA v4.3.3 [27]. To ensure removal of any human DNA contamination, non-rRNA reads were filtered using a two-step approach: (i) k-mer matching against a human pan-genome database with Kraken2 v2.0.8-beta [28,29] and (ii) alignment against the telmore-2-telomere reference genome (GCF_009914755) and human leukocyte antigen sequences (https://github.com/ANHIG/IMGTHLA/ URL accessed on 10 July 2024) using Bowtie2 [23].

These reads were de novo assembled using MEGAHIT v1.1.3 [30] and we remapped the filtered reads back to the resulting contigs using Bowtie2 [23] to extract unassembled reads. Kallisto v0.46.0 was used to establish contig abundance [31]. We then aligned the contigs and unassembled reads to the NCBI nucleotide database (nt, downloaded 27 June 2023) using BLAST+ v.2.11.0 [32] and to the NCBI non-redundant protein database (nr, downloaded 27 June 2023) using Diamond v0.9.32 [33].

### 2.4. Novel Astrovirus Characterisation

We retrieved a complete and near complete genome of astroviruses with low sequence identity to known sequences available in our nt and nr databases from the galah and king parrot libraries, respectively. To retrieve more coverage of the king parrot-associated astrovirus genome, we determined relevant genomic positions of available contigs in GeneiousPrime using known avastrovirus sequences as reference and remapped the host filtered non-rRNA reads to this draft genome using bbmap v37.98 [34]. We annotated both novel astrovirus genomes with canonical ORFs of the virus family and with protein domains according to conserved domain database search from NCBI [35,36] and HMMER [37,38].

### 2.5. Phylogenetic Analyses

We aligned the translated protein sequence of the RdRp domain of our novel viruses with all known astrovirus sequences available on GenBank (accessed 15 August 2024) using the L-INS-i algorithm in MAFFT v7.490 [39]. The alignment was trimmed manually and sites with >80% gaps were removed. The alignment was modified to display only representative astrovirus species with ≤95% amino acid identity in this domain to other available sequences. This alignment was used to generate a maximum likelihood phylogeny in IQ-TREE 2 v2.2.2 with 1000 bootstrap branch supports and the best fit substitution model selection [40]. This process was repeated using translated sequences in the CaP domain. A representative phylogeny, including the same accessions as the RdRp, was constructed, as well as a more extensive phylogeny to include richer diversity of ANV strains and passerine astroviruses. A raw amino acid alignment of the complete ORF2 was constructed in MAFFT with the E-INS-I algorithm to determine species demarcation according to ICTV guidelines.

### 2.6. Predictive Analyses

To investigate patterns of asparagine-linked glycosylation (N-glycosylation) on the capsid proteins of the novel viruses, we queried amino acid sequences of the entire ORF2 translation in the NetNGlyc-1.0 server prediction tool [41]. This analysis was expanded to include all sequences utilised in the expanded CaP domain phylogeny. Predicted N-glycosylation sites with significant NetNGlyc results (>0.5) were annotated on each sequence of the complete ORF2 alignment. The alignment was also annotated with the locations of the capsid precursor and spike domains of Turkey astrovirus 2 (TAstV-2), previously determined with X-ray crystallography [42], and ends were trimmed manually for visualisation.

To determine the predicted 3D structure of the novel spike proteins, we performed in silico protein homology modelling using SWISS-MODEL [43,44]. Protein translations of ORF2 were used as queries and models were built using template crystal structure of TAstV-2 (SMTL ID 3ts3.1). This analysis was then repeated for sequences that captured representative diversity of the genus to contextualise the findings of the novel viruses.

## 3. Results

### 3.1. Necropsy Findings

The galah and king parrot were found to have loops of small bowel that were thin-walled to the point of being nearly transparent, and that contained a liquid bile-stained content. The most significant microscopic lesions were a diffuse severe chronic lymphoplasmacytic enteritis with villus blunting and fusion and hyperplasia of the crypt epithelium of the duodenum and proximal jejunum. Renal, hepatic, and brain lesions were not observed.

### 3.2. Metatranscriptomic Libraries

Following quality trimming, host removal, and rRNA depletion, 9.33% (n = 5,630,332/60,338,927 read pairs) and 19.93% (n = 3,926,189/19,694,966 read pairs) of the galah and parrot libraries were available for assembly and taxonomic assignment, respectively. First, from the galah library, we recovered a genome-length astrovirus-like contig (6894 bp). Over 50% of the quality filtered, non-host non-rRNA read pairs in the galah library mapped to this sequence, with an abundance of 6.7 × 10^4^ transcripts per million (TPM). Next, from the king parrot library, we assembled three contigs into a genome-length scaffold (6825 bp) and finalised the genomic sequence through read mapping. The completed genome (6836 bp) had >2% of the quality filtered reads mapped to it. Based on the hosts in which they were identified, we propose naming these novel viruses *Avastrovirus eolorosei* (galah astrovirus, GAstV) and *Avastrovirus aliscap* (king parrot astrovirus, KPAstV), and their sequences have been deposited to GenBank under the accessions PQ893528 and PQ893527, respectively.

Beak-and-feather virus (BFDV) was identified at lower abundance (<0.1% of filtered reads) in both libraries. No other significant or potential avian viral pathogens were present in the metatranscriptomes. Assessment of the potential bacterial and eukaryotic pathogens in these libraries was outside the scope of this study.

### 3.3. Genome Annotation and Alignments

The genome architecture of both viruses is typical of astroviruses, with three large ORFs: ORF1a (GAstV, 3096 bp; KPAstV, 2901 bp), ORF1a (GAstV, 1524 bp; KPAstV, 1470 bp), and ORF2 (GAstV, 2037 bp; KPAstV, 2058 bp) (Figure 1). Both genomes include the heptameric slippery ribosomal frameshift motif “AAAAAAC” between ORF1a and ORF1b as well as the putative RNA synthesis promoter, which lies upstream of ORF2 [45]. Both viruses differ to the consensus sequence provided in [45] by only four 5′ end nucleotides each (Figure 1). KPAstV contains only four nucleotides in the 3′ variable region of this promoter, directly upstream of the ORF2 start codon, which, typically for avastroviruses, contains five to eight bases [45].

The highly conserved RdRp domains of GAstV and KPAstV have low protein identity (<65.2 and <83.5%, respectively) with publicly available astrovirus sequences as of August 2024 (Table 1, Appendix A). Similarly, a masked alignment of the capsid precursor domain revealed only 47.3 and 74.2% identity among closest relatives of these viruses, respectively. KPAstV shares maximum identity across both domains and the complete translated sequence of ORF2 with Avian astrovirus strain pigeon/China/20/2014 (MF768270). GAstV holds greatest identity with three different viruses across the three alignments, likely due to its deep divergence. Both viruses exceed the minimum requirements of species demarcation criteria outlined by the current report [3] of the International Committee for the Taxonomy of Viruses, having distinct hosts from their phylogenetic neighbours and <75.0% amino acid identity with any other avastrovirus in the complete ORF2 translation alignment, hence we suggest they both be considered novel *Avastrovirus* species.

### 3.4. Phylogenetic Analyses Reveal KPAstV and GAstV Are Divergent from Other Avastrovirus Species

Both KPAstV and GAstV reside within the broader diversity of the *Avastrovirus* genus (Figure 2). The discovery of these viruses in Psittaciformes expands the number of avian orders known to host astroviruses to eight. Phylogenetic analyses of both conserved domains revealed that KPAstV clusters within Clade 2 among ANV and avastroviruses from a wide range of unrelated avian hosts (Figure 2, Appendix A). The RdRp of GAstV, on the other hand, lies outside the known diversity of avian-hosted avastroviruses but its CaP clusters within poultry-associated avastroviruses of Clade 1 (Figure 2). The congruence in phylogenetic topology of the KPAstV domains suggests ancient recombination is unlikely. The relative novelty of the GAstV RdRp domain, compared to its CaP domain, may be explained by recombination but, without more closely related strains for comparison, this is impossible to assess.

### 3.5. Predicitive Structural Analyses Show Support for Phylogenetic Relationships

To compare the biology, evolution, and potential pathogenesis of avastroviruses, we examined the capsid and spike domains of ORF2 using structural and glycosylation prediction analyses. All clades of Avastrovirus show more confident, numerous, and conserved predicted N-glycosylation (GlyN) patterns in the conserved CaP domain than in the spike domain (Appendix A). The region with greatest N-glycosylation occurs in the CaP domain of Clade IV, the passerine astroviruses, followed by Clade III. KPAstV has two CaP-associated GlyNs but none in the spike domain, whereas GAstV has one in each. The first GlyN of KPAstV (N220 Y221 T222 G223) is identical to that of Neva virus (MT993583). Its second (N336 D337 T338 S339) has 75% identity to the Feral pigeon astrovirus (FR727146), N344 D345 T346 K347. The CaP GlyN of GAstV (N81 R82 S83 W84) has 75% identity with three crow-associated astroviruses (KT946719-20, MW385413). There is no single conserved GlyN in the spike domain of ANV species (n = 15) nor across the canonical poultry pathogens. In contrast, a highly conserved GlyN is present in the CaP domain at sites 319–322 of the ORF2 alignment across all four major clades of Avastrovirus—in nearly all ANV strains, a wood pigeon astrovirus, a red-necked avocet astrovirus, mute swan faeces-associated astroviruses, several passerine astroviruses, and the Hetplan gecko astrovirus.

Predicted protein models based on the experimentally derived crystal structure of TAstV-2 capsid spike [42] reveal two distinct structures within avastrovirus Clades I and II (Figure 3, Appendix A). Significant 3-dimensional homology with the homodimer configuration of TAstV-2, with centrally aligned N- and C-termini, asymmetrical apical bulges, and surface glycosylation, is observed in other representatives from Clade I (Figure 3L,M) and in three strains of ANV (Figure 3H–J). It is also the predicted structure of Neva virus and 2a feral pigeon avian astrovirus from Clade II (Figure 3C,F). These are outliers among other members of the clade, where KPAstV, two pigeon astroviruses, and two red-necked avocet astroviruses share a distinct structure that features symmetrically distributed dimers, surface hairpins, weaker central alignment of the termini, and little to no surface glycosylation (Figure 3A,B,D,E,G,N). Sequences that aligned with the TAstV-2 structure had significantly higher amino acid percentage identity with TAstV-2, Global Model Quality Estimates (GMQE), and QMEANDisCo global scores than sequences with KPAstV-like models (Appendix A). No templates were found for the C-terminal region of the GAstV ORF2 protein, nor of Astroviridae sp. (Clade II, MT138010), Hetplan gecko astrovirus (Clade III, PP711188), and Passerine astrovirus 1 (Clade IV, MK096773), so were not included further in the analysis. Together, these results show that there was no apparent conserved spike structure within the KPAstV-related avastrovirus clade.

## 4. Discussion

Our use of unbiased metatranscriptomics to identify viruses suspected to be associated with the pathogenesis of a wasting disease in several species of Australian parrots identified two highly abundant novel avastroviruses in the digestive tract of two species of parrots with this syndrome. These findings are significant because they contribute to the understanding of avastrovirus evolution by expanding the diversity and host range of the genus, providing the first two astroviruses infecting psittacine birds. It also provides potential insights into the pathogenesis of a significant wildlife disease.

The genomic architecture and identities of GAstV and KPAstV places them as novel proposed species within the genus *Avastrovirus* (Figure 1, Table 1). The deep divergence of the RdRp in GAstV from other avastroviruses suggests that the range of genetic diversity in the genus is greater than previously defined. KPAstV adds to the host complexity of its diverse clade, which includes both pathogenic (e.g., ANV, pigeon-associated avastroviruses) and non-pathogenic (e.g., ruddy turnstone astroviruses) members [5,6,7,46]. The recent discoveries of avastro-like viruses in lizards and geckos further highlights the expanding genetic diversity of this genus (Figure 2) [2,47]. Additionally, the discrepancy in the RdRp and CaP tree topologies could be evidence of ancient recombination events within this unexplored diversity. Despite sampling only two individuals, the deep divergence between KPAstV and GAstV supports the notion that wild birds may harbor a vast, unidentified reservoir of enzootic viruses [4]. Avastroviruses can evolve through host co-speciation or cross-species transmission, as seen in ANV and turkey astroviruses (Table 1, Figure 2) [6,9,45,48,49]. However, divergent avastroviruses (e.g., Carnarvon virus, Blencathra virus, and Neva virus) have been identified in single populations of a single host species (Figure 2) [5]. Therefore, the origins of KPAstV and GAstV remain unclear. Resolving these host-virus dynamics requires expanded metatranscriptomic surveys of Australian avifauna, sampling both healthy and diseased wild birds.

N-linked glycosylation of viral proteins enhances infectivity, immune evasion, and host specificity in RNA viruses like avian influenza and coronaviruses [50,51,52,53]. However, its role in avian astrovirus capsids remains unexplored, though human astrovirus capsid maturation and cleavage are well documented [42,54,55,56]. We compared ORF2 N-glycosylation sites to assess host specificity and pathogenicity, presenting the first comparative analysis of these patterns across avastroviruses (Appendix A). We identified two well-supported N-glycosylation sites in the KPAstV capsid precursor, similar to other Clade II viruses. In contrast, GAstV showed no resemblance to any avastrovirus clade. While Clades III and IV exhibited some grouping in the capsid precursor, spike protein glycosylation showed little conservation. Given its importance in other RNA viruses, future research on capsid and spike proteins could enhance pathogenicity assessments as more avian and mammalian astroviruses emerge.

Predicted spike protein models offer insights into Clade I and II avastrovirus evolution and host specificity (Figure 3). The TAstV-2 crystal structure showed only distant similarity to human astrovirus spikes, suggesting an avian-specific structure [42]. Despite <30% protein identity among avastrovirus spikes, our analysis identified two distinct structures: one in poultry-borne Clade I and ANV strains, and another in wild bird-borne Clade II, with exceptions in Neva virus and AAstV-fp (Figure 3, Appendix A). This may reflect convergent evolution or an ancestral phasianid-specific trait. However, predictions are limited by access to only one experimentally derived protein crystal structure, despite the avicultural significance of this genus [42]. Determining the ANV spike structure, given its phylogenetic distinction and relevance as a model, would clarify its and its relatives’ roles in host interactions and virulence. This work highlights the potential value of analysing the spike protein structure to assess the long-term evolution of avastroviruses across the broad range of hosts they infect.

Astroviruses can cause acute or chronic diarrhoea, enteritis, and other diseases. Enteric astrovirus infections are typically acute and self-limiting [13,57,58]. In birds and mammals, they infect mature enterocytes at the villous tips of the small intestine, leading to villous atrophy and epithelial damage. This disruption of absorptive function results in malabsorption and osmotic diarrhoea. Concurrently, barrier dysfunction and immune activation induce mild inflammation that collectively aggravate the diarrhoea and enteritis. The severe chronic lymphoplasmacytic enteritis and blunting and fusion of the villi in our two cases are consistent with histopathology observed in acute astrovirus infections in turkey, chickens, geese, ducks, and pigeons [6,45,46,59,60,61,62], and hence may have been an important contributor to the diarrhoea and weight loss seen in these birds and other birds with this syndrome.

The chronic disease presentation and progression in the two case individuals is very similar to those of galahs and a sulfur-crested cockatoo from Perth [18], galahs and sulfur-crested cockatoos from Victoria in 1991 [19], king parrots from the south-eastern states of Australia through 1984–2000 [20], and galahs in south-east Queensland from 2002–2012 [21]. It is also similar to what has been observed in immunosuppressed humans [57,63]. Concurrent infections with the immunosuppressive psittacine beak and feather disease virus (BFDV) [64,65] were observed in the two birds in this study and has been documented in other parrots with similar presentations [18]. Co-infection with BFDV may be an important factor in the chronic nature of these avastrovirus infections. Further study is required to elucidate the specific roles of these viruses in the aetiology of this chronic psittacine wasting disease [18,19,20,21].

Based on the metatranscriptomic abundance of astrovirus in both intestinal libraries, the two parrots in this study may have been shedding large concentrations of these novel viruses. We thus propose that birds with this syndrome should be strictly quarantined when hospitalised or housed in rehabilitation facilities so as to not potentially infect other birds. Additionally, Australia contains one of the largest groups of endemic parrot species and, consequently, there are considerable biosecurity concerns that importation of exotic parrots either for pets or for aviculture may introduce pathogens that are not present in Australia. The 2020 Australian Government report on psittacine bird imports deemed astroviruses as “not of significance” [66]. Our work demonstrates that novel avastroviruses can infect parrots. Therefore, psittacine birds being imported into Australia should be tested for and be free of avastroviruses before they are allowed to enter the country.

## Figures and Tables

**Figure 1 viruses-17-00450-f001:**
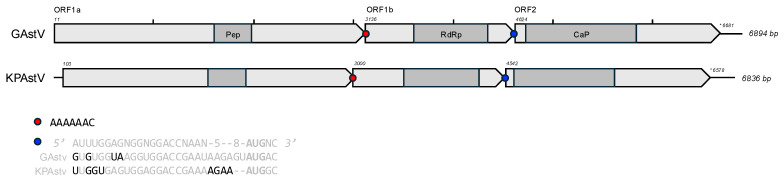
Genome architecture of GAstV (PQ893528) and KPAstV (PQ893527). Both viruses contain the canonical three open reading frames of astroviruses, shown in light grey arrows. Conserved protein domains are depicted in dark grey include the trypsin-like peptidase (pfam13365), Astroviridae RNA-dependent RNA polymerase (cd23172), and Astrovirus capsid protein precursor (cl46880). The nucleotide position of each start codon is provided above each sequence, as well as the coordinates of the ORF2 stop codons. Tick marks indicate 1000 bp intervals and the complete genome length is provided at the end of each sequence. Red circles mark the ribosomal slippage heptamer between ORF1a and ORF1b. Blue circles indicate the site of the subgenomic RNA synthesis promoter sequence for ORF2. An alignment of the novel species subgenomic ORF2 promoters with the consensus sequence provided in [45] is included. Deviations from the proposed promoter are shown in black letters and the ORF2 start codon is indicated in bold.

**Figure 2 viruses-17-00450-f002:**
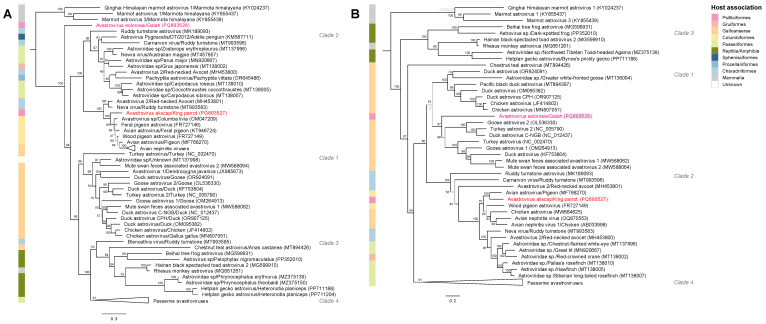
Maximum likelihood phylogenies of avian astroviruses. (**A**) The RdRp tree was generated from an alignment of 250 amino acid residues of the *Astroviridae* family RdRp domain (cd23172). (**B**) The CaP tree was generated from an alignment of 377 amino acid residues of the astrovirus capsid precursor protein domain (cl46880). Phylogenies were constructed in IQ-TREE 2 [40] using the LG model with 1000 Ultrafast bootstrap supports (shown at nodes out of 100) and are rooted on the marmot astrovirus outgroup. Nodes with fewer than 70% support are drawn in pale grey. GAstV and KPAstV are shown in pink and red text, respectively. Phylogenetic clades that are consistent across both domains are shaded with grey boxes. Host association for each sequence is summarised in panels adjacent to each tree with colours representing avian orders and non-avian classes: Reptilia and Amphibia (dark green), Mammalia (grey), Anseriformes and Galliformes (orange), Charadriiformes (pale blue), Columbiformes (yellow), Gruiformes (pale red), Passeriformes (pale green), Procellariiformes (blue), Psitticiformes (pink) and Sphenisciformes (dark blue).

**Figure 3 viruses-17-00450-f003:**
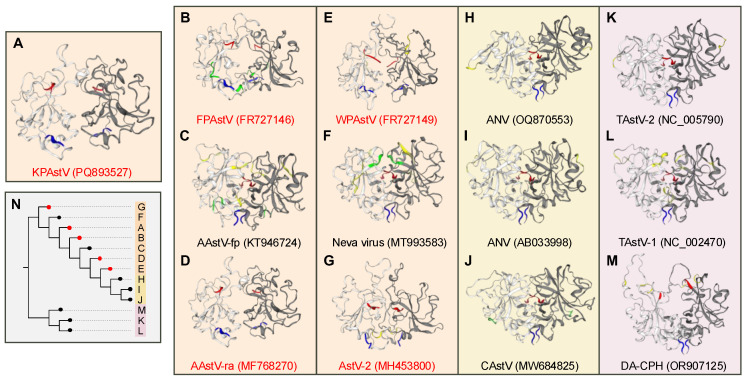
Predicted homo-dimeric protein structures of avastrovirus capsid spike domains. C-terminus regions of translated ORF2 beyond the conserved capsid precursor protein domain were searched against the database of protein structure templates in SWISS-MODEL. Models were built where homology to the Turkey astrovirus 2 capsid spike 3ts3.1.A template [42] was found in (**A**) *Avastrovirus aliscap* (KPAstV), king parrot, (**B**) *Avian astrovirus* (AAstV-ra), pigeon, (**C**) *Wood pigeon astrovirus* (WPAstV), wood pigeon, (**D**) *Feral pigeon astrovirus* (FPAstV), feral pigeon, (**E**) *Neva virus*, ruddy turnstone, (**F**) *Avastrovirus 2* (AstV-2), red-necked avocet, (**G**) *Avian astrovirus* (AAstV-fp), feral pigeon, (**H**) *Avian nephritis virus* (ANV), chicken, (**I**) *Avian nephritis virus* (ANV), chicken, (**J**) *Chicken astrovirus* (CAstV), chicken, (**K**) *Turkey astrovirus 2* (TAstV-2), turkey, (**L**) *Turkey astrovirus 1* (TAstV-1), and (**M**) *Duck astrovirus CPH* (DA-CPH), duck. (**N**) The relative phylogenetic relationships between the viruses analysed in A-M as a cladogram. Chains of each homodimer are drawn in white and grey, with N- and C-termini drawn in blue and red, respectively. Predicted N-glycosylation tetrapeptides are drawn in yellow (N-Glyc result 0.5–0.65) and green (N-Glyc result 0.65–0.75). Background colours (panels A–M) and tip label highlighting (panel N) correspond to Clade I (pink), non-ANV Clade II (orange), and ANV Clade II (yellow) from Figure 2. Dimeric structures similar to KPAstV are written (panels A–M) and depicted with tip circles (panel N) in red, while those similar to the crystal structure of TAstV-2 are labelled in black.

**Table 1 viruses-17-00450-t001:** Mean percent identity of GAstV and KPAstV with other avastrovirus groups. Protein identity was provided alignments of each region. Domain-based alignments had sites containing greater than 80% gaps stripped. The mean was calculated across consistent phylogenetic clades defined in Figure 2. The maximum identity score and reference accession is provided for each novel virus and alignment. The alignments upon which these scores were calculated are available in the Appendix A.

Alignment	Mean ± SD %ID	GAstV	KPAstV
RdRp domain	Clade 1 (poultry)	61.68 ± 1.59	58.10 ± 2.06
	Clade 2 (diverse)	61.68 ± 3.13	71.41 ± 9.26
	Clade 3 (herp)	51.37 ± 5.14	48.82 ± 3.14
	Clade 4 (passerine)	59.61 ± 1.83	55.09 ± 1.66
	Outgroup (marmot)	46.36 ± 1.52	45.45 ± 0.23
	*Maximum ID*	65.2 ^a^	83.53 ^b^
Capsid precursor domain	Clade 1 (poultry)	39.30 ± 2.99	38.18 ± 1.94
	Clade 2 (diverse)	46.11 ± 1.62	37.93 ± 8.22
	Clade 3 (herp)	31.33 ± 1.83	35.28 ± 2.28
	Clade 4 (passerine)	36.82 ± 1.86	44.76 ± 1.83
	Outgroup (marmot)	27.09 ± 1.03	25.46 ± 2.18
	*Maximum ID*	47.28 ^c^	74.22 ^b^
Complete ORF2	Clade 1 (poultry)	28.81 ± 3.71	24.09 ± 4.03
	Clade 2 (diverse)	22.79 ± 1.00	46.26 ± 08.28
	Clade 3 (herp)	19.39 ± 5.66	19.44 ± 7.47
	Clade 4 (passerine)	19.78 ± 3.08	18.29 ± 3.54
	Outgroup (marmot)	7.75 ± 0.48	6.71 ± 0.46
	*Maximum ID*	35.17 ^d^	58.57 ^b^

^a^ MT138004, ^b^ MF768270, ^c^ MW588064, ^d^ MT894397.

## Data Availability

Quality- and host-filtered paired reads have been deposited in the Sequence Read Archive (SRA) under the accession numbers SRR31990207-8. The genomic sequences of KPAstV and GAstV are available on GenBank under the accession numbers PQ893527 and PQ893528, respectively. Appendix A including raw and masked alignments and PDB files for protein models are available on GitHub: https://github.com/jsede/Avastro (URL accessed 10 March 2025).

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
