# Peer review of "Two Avastrovirus Species Discovered in Psittaciformes Expand the Host Range of the Family Astroviridae"

_viruses, 2025, doi:10.3390/v17030450_

Round 1

Reviewer 1 Report

Comments and Suggestions for Authors

The manuscript titled “Two novel avastrovirus species expand the host range of Astroviridae to Psittaciformes” addresses the identification astroviruses in parrots.

I do not have major comments for the authors. The MS is straight forward with the identification of the two genomes via metatranscriptomic approaches, and the homology modelling of the spike protein coupled with prediction of glycosylated residues on the surface of the spike.

Minor comments

  • I would suggest rewriting the first sentence of the abstract “Metatranscriptomics has recently expanded the species richness and host range of the 11 Avastrovirus genus, quadrupling the number of avian orders known to host them in less than a decade.” It is via the use of metatranscriptomic that researchers have been able to determine a broader host range and species richness … Please note that these viruses already existed in nature – they just were not identified prior to this study.
  • Line 87 and elsewhere. We have moved past next generation sequencing, we are already in the 3rd generation – single molecule sequencing. Rather just state high throughput sequencing.
  • Line 144: please change x-ray diffraction to x-ray crystallography which is the correct term for this technique of solving protein structures at atomic resolution
  • Line 137: please define KPAstV and GAstV and this is the first use of these acronyms.
  • 167: Virus species names are not abbreviated and as it stands the abbreviations don’t reflect the species names - Avastrovirus eolorosei (GAstV, from the galah) and Avastrovirus aliscap (KPAstV, from the king parrot). I think what you are trying to say here is that GAstV and KPAstV are members of two new species Avastrovirus eolorosei and Avastrovirus aliscap.
  • Figure 2: it would really help the reader if you can show the branch support and also if you can have a colour key in the figure in addition to what you have in the figure legend “Host association for each sequence is summarised in panels adjacent to each tree with colours representing avian orders and non-avian classes: Reptilia and Amphibia (dark green), Mammalia (grey), Anseriformes and Galliformes (orange), Charadriiformes (pale blue), Columbiformes (yellow), Gruiformes (pale red), Passeriformes (pale green), Procellariiformes (blue), Psitticiformes (pink) and Sphenisciformes (dark blue).”

Author Response

I would suggest rewriting the first sentence of the abstract “Metatranscriptomics has recently expanded the species richness and host range of the 11 Avastrovirus genus, quadrupling the number of avian orders known to host them in less than a decade.” It is via the use of metatranscriptomic that researchers have been able to determine a broader host range and species richness … Please note that these viruses already existed in nature – they just were not identified prior to this study.

We have changed the sentence to: “Metatranscriptomics has recently revealed greater species richness and host range of the 11 Avastrovirus genus, quadrupling the number of avian orders known to host them in less than a decade.” Please see revised manuscript line 11-12.

Line 87 and elsewhere. We have moved past next generation sequencing; we are already in the 3rd generation – single molecule sequencing. Rather just state high throughput sequencing.

We have made changes accordingly. Please see revised manuscript lines 81, 88.

Line 144: please change x-ray diffraction to x-ray crystallography which is the correct term for this technique of solving protein structures at atomic resolution

Thank you, we have changed this accordingly. Please see revised manuscript line 146.

Line 137: please define KPAstV and GAstV and this is the first use of these acronyms.

Done. Please see comment 5 below for further detail.

Line 167: Virus species names are not abbreviated and as it stands the abbreviations don’t reflect the species names - Avastrovirus eolorosei (GAstV, from the galah) and Avastrovirus aliscap (KPAstV, from the king parrot). I think what you are trying to say here is that GAstV and KPAstV are members of two new species Avastrovirus eolorosei and Avastrovirus aliscap.

Thank you for pointing out this needs clarifying. We have changed the manuscript to exclude any specific naming in the methods, just referring to “the novel viruses” to allow for introduction of the names and abbreviations in the results (revised manuscript lines 118-9, 125, 140, 148). The rationale behind our naming system follows ICTV’s recent incentive to give virus species binomial taxonomic names. Therefore, we name our virus species Avastrovirus eolorosei (taking “eolo” and “rosei” from the binomial name of the galah, Eolophus roseicapilla) and A. aliscap (similarly, taking “ali” and “scap” from Alisterus scapularis, king parrot). But for ease of reference and naming we also provide common names, Galah astrovirus and King parrot astrovirus, and make abbreviations based on these – GAstV and KPAstV, in similar custom to other astrovirus abbreviations e.g., Human astroviruses (HAstVs), Porcine astrovirus (PAstV). Please see revised manuscript lines 166-8 for introduction of the proposed virus names.

Figure 2: it would really help the reader if you can show the branch support and also if you can have a colour key in the figure in addition to what you have in the figure legend “Host association for each sequence is summarised in panels adjacent to each tree with colours representing avian orders and non-avian classes: Reptilia and Amphibia (dark green), Mammalia (grey), Anseriformes and Galliformes (orange), Charadriiformes (pale blue), Columbiformes (yellow), Gruiformes (pale red), Passeriformes (pale green), Procellariiformes (blue), Psitticiformes (pink) and Sphenisciformes (dark blue).”

Thank you, we have added a legend and bootstrap values (out of 100) to the trees in Figure 2. Please see revised manuscript line 230 and 235.

Reviewer 2 Report

Comments and Suggestions for Authors

The study of Jenns and co-workers describes the NGS-based detection and phylogenetic/sequence analyses of complete/nearly complete genomes of two novel astroviruses (GAstV and KPAstV) from intestinal samples of a diarrheic galah (Eolophus roseicapilla) and an Australian king parrot.

Based on the results of phylogenetic and sequence analyses, both viruses could be founding members of two novel avastrovirus genotype species.

The findings are somewhat interesting, but the manuscript in present state is too long and did not contain enough results to justify the format (research article), it should be rewritten as short/brief communication.

Major points

(i) The whole discussion should be more concise. The authors should discuss the results only and should not wander far away from the subject. e.g. lines 303-335 and lines 394-400 are only loosely connected to the results and the subject of the MS.

(ii) There is no convincing evidence which supports the role of detected AstVs in the manifested symptoms (severe pectoral muscle atrophy/weight loss and diarrhea). The authors did not conduct any epidemiological investigations (asymptomatic vs. diseased) of these viruses (although multiple samples were available at least from galahs, see line 59) which could support this assumption. The authors suggest a pathogenic role of these AstVs solely based on the predominance of related reads in NGS data (over >50%, line 162), and beside Beak-and-feather virus they did not share any information about the classification of the other non-AstV, non-BFDV reads.

(iii) The authors state that “…detect the entire genomes of two novel astrovirus species, Avastrovirus eolorosei (PQ893528) and Avastrovirus aliscap (PQ893527).” (in lines 16-17) and … we describe the complete sequences of two novel avastroviruses… (in line 44), but according to the GenBank flatfile of KPAstV QLD/2021 (PQ893527), it is not complete, its 5’ end is missing, including the 5’ UTR and the start of ORF1a. The authors should clearly indicate throughout the manuscript as well as in figure 1 which sequence is complete and which is partial, and which region is missing.

(iv): Naming an AstV species is the responsibility of the ICTV. Please use the word “proposed” when the names of the study viruses are introduced (e.g. lines 17, 167). Did the author submit an official proposal about the species names for ICTV?

(v): The species demarcation criteria of AstV genotype species are based on the comparisons of full-length amino acid sequences, not “manually trimmed/ sites with >80% gaps were stripped, and masked alignments” (lines 126-127, 191, 202). The sequence identity values of the full-length capsid and NS of study AstVs should be provided at least with the closest known relative(s).

(vi): Without the access of the “masked alignments” used for identity calculations the data presented in Table 1 is meaningless. The pairwise identity values should be calculated from the full-length ORF1a/1b and ORF2 alignments.

Minor points

Line 26: ORF1a is encoding not just a serine protease

Line 137: The full names of the viruses should be included here right before the first appearance of the abbreviations.

Line 163: What does “TPM” means?

Lines 173-179: Lengths of the genomic sequences should be included here: e.g. length of the ORF1a/b, ORF2.

Line 181 (Figure 1): A ruler above the genome maps should be included. nt positions of the start and stop codons also should be included (should be indicated here the missing 5’ end of KPAstV), including the positions of the indicated nt motifs. What is the first/top sequence marked with a blue dot? (what is a proposed subgenomic RNA synthesis promoter sequence? Where did it come from?) Genomic positions of the first and last nt of GAstV and KPAstV marked with a blue dot should be included.

Line 177: Family name (Astroviridae) should be written in italics

Lines 209-210: Please consider moving this sentence to Discussion.

Line 215: Details of the recombination analyses are completely missing.

Line 219 (Figure 2): The names and locations of official Avastrovirus genotype-species should be included in the phylogenetic trees.

Line 426: The accession numbers of the AstV sequences should be included here as well

Line 397: Without quantitative measurement (i.e. RT-qPCR or Digital PCR) such expressions should be avoided.

Lines 398-400. This section is out of the scope of this study. It should be omitted from the Discussion.

Lines 411-412.: There is no evidence which supports this hypothesis. It should be omitted from the Discussion.

Author Response

The findings are somewhat interesting, but the manuscript in present state is too long and did not contain enough results to justify the format (research article), it should be rewritten as short/brief communication.

We have made efforts to reduce the article length substantially to improve clarity and focus, yet believe that these are important findings to share in a research article format.

Major points

The whole discussion should be more concise. The authors should discuss the results only and should not wander far away from the subject. e.g. lines 303-335 and lines 394-400 are only loosely connected to the results and the subject of the MS.

Thank you for your recommendation. We have condensed the discussion from three to two pages long, with our changes indicated by red text in the revised manuscript between lines 299-381.

There is no convincing evidence which supports the role of detected AstVs in the manifested symptoms (severe pectoral muscle atrophy/weight loss and diarrhea). The authors did not conduct any epidemiological investigations (asymptomatic vs. diseased) of these viruses (although multiple samples were available at least from galahs, see line 59) which could support this assumption. The authors suggest a pathogenic role of these AstVs solely based on the predominance of related reads in NGS data (over >50%, line 162), and beside Beak-and-feather virus they did not share any information about the classification of the other non-AstV, non-BFDV reads.

The aim of our paper is to introduce and characterise these novel astroviruses. Please see revised manuscript lines 175-6 for clarification about the remaining reads. Unfortunately, we did not have access to the galah specimens to which you refer. We have attempted to provide as much information about the cases we present, which we believe are important as the first reports of psittacine avastrovirus infection. While we show that the birds have histology and disease progression that could be consistent with an astrovirus infection as seen in other birds [6,45,51,63-66], we try to only suggest that the viruses we found may be associated with disease. We have now added references to support the claim that BFDV causes immune suppression [68-69]. We do not suggest that BFDV causes the signs that we report, instead we suggest that it just may be important to the chronic nature of the syndrome and requires further study. Please see revised manuscript lines 365-370. We hope this clarifies your concerns.

The authors state that “…detect the entire genomes of two novel astrovirus species, Avastrovirus eolorosei (PQ893528) and Avastrovirus aliscap (PQ893527).” (in lines 16-17) and … we describe the complete sequences of two novel avastroviruses… (in line 44), but according to the GenBank flatfile of KPAstV QLD/2021 (PQ893527), it is not complete, its 5’ end is missing, including the 5’ UTR and the start of ORF1a. The authors should clearly indicate throughout the manuscript as well as in figure 1 which sequence is complete and which is partial, and which region is missing.

All avian astroviruses available on GenBank, when analysed with an ORF-finding program show open reading frames all the way to the 5’ terminus. Some authors have annotated their viruses with ORF1a CDSs as truncated. Others ignore the fact that the reading frame remains open through to the 5’ terminus of their sequences and instead mark the ‘start’ of their ORF1a at the first methionine residue. In our initial submission, we were following the former convention. We have now updated our annotations on GenBank, and in Figure 1, to reflect the latter convention – choosing the first methionine residue to start ORF1a. Please note, we can find little to no evidence of any published literature experimentally confirming the true start codons of ORF1a in any avastrovirus species, so if you accept the convention of others annotating from the first methionine residue’, we will proceed accordingly. We re-interrogated our data for the king parrot library and have updated an additional 11 bp to the 5’ of this sequence. We are confident that this genome is complete, and the 5’ UTR is not missing. Please see lines 166-168 in the revised manuscript for clarification and an updated version of Figure 1, showing the 5’ of the KPAstV.

Naming an AstV species is the responsibility of the ICTV. Please use the word “proposed” when the names of the study viruses are introduced (e.g. lines 17, 167). Did the author submit an official proposal about the species names for ICTV?

We have modified the manuscript to emphasise that the species names are proposed. Please see lines 166-168 in the revised manuscript to see our naming proposals and abbreviations. We have aimed to satisfy the ICTV’s recent transition towards providing viruses with binomial Latin species names, in addition to and/or in replacement of common names. We provide proposed binomial and common names for these novel species, for ease of communication.

The species demarcation criteria of AstV genotype species are based on the comparisons of full-length amino acid sequences, not “manually trimmed/ sites with >80% gaps were stripped, and masked alignments” (lines 126-127, 191, 202). The sequence identity values of the full-length capsid and NS of study AstVs should be provided at least with the closest known relative(s).

We specify that only the domain-based alignments were trimmed and masked. Please see lines 135-137, 212, 214 for clarification. Closest known relative identity for the capsid and NS are provided in Table 1, please see lines 217 of the revised manuscript.

Without the access of the “masked alignments” used for identity calculations the data presented in Table 1 is meaningless. The pairwise identity values should be calculated from the full-length ORF1a/1b and ORF2 alignments.

We have uploaded raw and trimmed/stripped alignment sequences to a GitHub repository that can be accessed in the “Data Availability” section of the revised manuscript. We refer to this as “Supplementary Data” in the main text (see lines 201, 215 of revised manuscript).

Minor points

Line 26: ORF1a is encoding not just a serine protease

The serine protease is the only domain of ORF1a that has been experimentally confirmed in astroviruses. Please see revised manuscript lines 26-27 where we have clarified.

Line 137: The full names of the viruses should be included here right before the first appearance of the abbreviations.

Based on another reviewer’s comments, we have omitted any naming of the novel viruses in the methods to introduce the names properly in the results. Please see revised manuscript lines 118-9, 125, 140, 148.

Line 163: What does “TPM” means?

Transcript per million – a common abundance metric in metatranscriptomic analyses. We have now included this in the revised manuscript, see line 166.

Lines 173-179: Lengths of the genomic sequences should be included here: e.g. length of the ORF1a/b, ORF2.

Please see revised Figure 1, line 187, reworked to include your comments, thank you.

Line 181 (Figure 1): A ruler above the genome maps should be included. nt positions of the start and stop codons also should be included (should be indicated here the missing 5’ end of KPAstV), including the positions of the indicated nt motifs. What is the first/top sequence marked with a blue dot? (what is a proposed subgenomic RNA synthesis promoter sequence? Where did it come from?) Genomic positions of the first and last nt of GAstV and KPAstV marked with a blue dot should be included.

Thank you for this feedback, the figure and legends has been modified according to your comments. Please see revised manuscript lines 187-198. We feel there is no need to included nucleotide positions of the ORF1a and ORF1b stop codons because the stop codon of the ORF1a is so close to the ORF1b start codon (ribosomal slippage). This is similar between ORF1b and ORF2 where there are fewer than 20 bp between the ORFs. We feel that adding the stop codons does not add any value to this figure and instead would clutter it and reduce the impact of its purpose, which is to provide a quick, simple overview of the structures. For readers who are interested in finer detail, they may consult the sequence submitted to GenBank.

Line 177: Family name (Astroviridae) should be written in italics

Please see revised line 184.

Lines 209-210: Please consider moving this sentence to Discussion.

We believe the topology of the novel viruses in context of other viruses in the phylogeny is a result, rather than discussion.

Line 215: Details of the recombination analyses are completely missing.

There was not enough data to perform recombination analyses, which is typically performed on very closely related virus species. These viruses are divergently related and other evolutionary processed might be topological incongruence. We can only make inferences on the potential recombination having occurred based on the topology to the two domain’s phylogenies. The topology of the KPAstV CaP and RdRp are largely congruent, suggesting a lack of recombination. However, there simply are not enough data for more closely related strains and species, especially for GAstV to even conduct meaningful recombination analyses. We have edited the manuscript to clarify this. Please see lines 226-229.

Line 219 (Figure 2): The names and locations of official Avastrovirus genotype-species should be included in the phylogenetic trees.

We chose not to include these as there has not been an update in Avastrovirus taxonomy for more than a decade and there has been a massive expansion of the genus that we believe requires revision from the ICTV. The authors do not wish to provide information on just “official Avastrovirus” genotype species where there has been 10-fold increase in the number of proposed species since, that remain officially unclassified, in anticipation of the current number of “official Avastrovirus” genotype species becoming obsolete or different in the near future.

Line 397: Without quantitative measurement (i.e. RT-qPCR or Digital PCR) such expressions should be avoided.

We have updated this in the manuscript accordingly to (line 371-373). “Based on the metatranscriptomic abundance of astrovirus in both individuals, the two parrots in this study may have been shedding large concentrations of these viruses.”

Lines 398-400. This section is out of the scope of this study. It should be omitted from the Discussion.

Our findings indicate that this disease syndrome may be viral in nature and thus is important to inform wildlife hospital practices. We have modified the language to acknowledge a potential, rather than confirmed, risk (lines 373-375).

Lines 411-412.: There is no evidence which supports this hypothesis. It should be omitted from the Discussion.

We have modified the language to remain within the bounds of our findings. Please see lines 379-381 in the revised manuscript.

Line 426: The accession numbers of the AstV sequences should be included here as well

Please see lines 394-395 in the revised manuscript for changes.

Reviewer 3 Report

Comments and Suggestions for Authors

In this article, authors present two newly sequenced genomes (one complete and one nearly complete) of novel astroviruses. A key strength of this study is that the viral sequences were obtained from two deceased birds with well described and documented disease symptoms, this is valuable context for the findings. Therefore, the study is undoubtedly worthy of publication, and our research group (in Russia) is prepared to cite this work in our nearest upcoming publications. However, I have one significant critique: the article is written in an overly complex and cumbersome style. The text is difficult to understanding and requires scientific editor|project curator|PI revision.

The authors omit logical connections, forcing the reader to re-read paragraphs multiple times to understand the underlying backgroung. 

For example, in line 160
"ibraries contained 5,630,332 (4.67% total) and 3,926,189 (9.97% total)"
What do the percentages in parentheses represent? Are they a percentage of the original number of reads obtained for the library?
To understand this, you have to stop reading the results and go back to the materials and methods. And still the numbers don't add up.

For example, in line 232, the paragraph begins with:
"To compare the biology, evolution, and potential pathogenesis of these viruses, we examined the capsid and spike domains of ORF2 using structural and glycosylation prediction analyses."

Here, the authors do not specify which viruses' capsid and spike domains were analyzed. Were these domains obtained from the viruses they sequenced only, or did they also include data from GenBank? While it becomes clear later in the paragraph that external data have been used, the exact sources and scope of the analysis remain unclear.

Additionally, the text contains incorrect use of terms, which decrease its readability. 

For example, in lines 220-221

"The CaP tree was generated from an alignment of 377 amino acids of the astrovirus capsid precursor protein domain (cl46880)"
Are authors meant to say ‘377 amino acid sequences of ?? length’ or ‘?? sequences of 377 a.a. length’? 
There is a big difference between the first and second statement, isn't?

In summary, I strongly recommend that the authors engage a scientific proofreader to revise the manuscript. This will ensure that the text is clear and enhancing the article's accessibility to the scientific community.

Comments on the Quality of English Language

I wrote in the main part of the review ‘the article is written in too complicated and cumbersome style’.
I am a foreigner, but I know English quite well. I read scientific articles in English every day, but it was difficult to read this manuscript. I recommend to improove text, linguistically it should be made clearer.

Author Response

The article is written in an overly complex and cumbersome style. The text is difficult to understanding and requires scientific editor|project curator|PI revision. The authors omit logical connections, forcing the reader to re-read paragraphs multiple times to understand the underlying backgroung. 

We have made a significant effort to reduce the complexity and length of the manuscript – most especially, in the discussion where other reviewers have focussed their comments. We believe the text is now clearer and more accessible. Please see revised manuscript lines 306-348, for example and the extensive edits in discussion (colored red).

For example, in line 160 "libraries contained 5,630,332 (4.67% total) and 3,926,189 (9.97% total)"
What do the percentages in parentheses represent? Are they a percentage of the original number of reads obtained for the library?
To understand this, you have to stop reading the results and go back to the materials and methods. And still the numbers don't add up.

Thank you, in addition to the missing denominator we did identify some errors as picked up on by the reviewer. This has all been corrected and full requested details provided. Please see revised manuscript lines 162-163.

For example, in line 232, the paragraph begins with:
"To compare the biology, evolution, and potential pathogenesis of these viruses, we examined the capsid and spike domains of ORF2 using structural and glycosylation prediction analyses." Here, the authors do not specify which viruses' capsid and spike domains were analysed. Were these domains obtained from the viruses they sequenced only, or did they also include data from GenBank? While it becomes clear later in the paragraph that external data have been used, the exact sources and scope of the analysis remain unclear.

This sentence is employed to guide the reader into a new topic without overloading on details. We believe the way it is phrased provides an aim and a summary of the approach taken, which is customary of most methods paragraphs’ opening sentences. The non-specific approach is required as we only analysed a subset of Avastrovirus species structural predictions. This is different from the glycosylation analyses, which was performed on all species analysed by phylogenetics earlier. GenBank accessions of sequences analysed are provided in Figure 3, underneath each model, on the left-hand side of Supplementary Figure 2, and the second column of Supplementary Table 1. Please see revised manuscript lines 244 and 263 for clarification.

Additionally, the text contains incorrect use of terms, which decrease its readability. For example, in lines 220-221 "The CaP tree was generated from an alignment of 377 amino acids of the astrovirus capsid precursor protein domain (cl46880)"
Are authors meant to say ‘377 amino acid sequences of ?? length’ or ‘?? sequences of 377 a.a. length’? There is a big difference between the first and second statement, isn't?

We have updated the figure legend to refer to “amino acid residues” which is customary of phylogenetic tree figure legends. The number of sequences used in the alignment is apparent as the number of tips on the tree. Please see revised manuscript line 230-242.

In summary, I strongly recommend that the authors engage a scientific proofreader to revise the manuscript. This will ensure that the text is clear and enhancing the article's accessibility to the scientific community.

We have made a significant effort to reduce the complexity and length of the manuscript and all authors have thoroughly proof-read the manuscript to ensure it is of a high academic standard.

Reviewer 4 Report

Comments and Suggestions for Authors

In this study, the authors performed RNA sequencing analysis of intestinal samples from Galahs (Eolophus roseicapilla) and Australian king parrot (Alisterus scapularis) suffering from chronic diarrheal wasting disease in Australia, and detected the whole genomes of two new astroviruses, Avastrovirus eolorosei (PQ893528) and Avastrovirus aliscap (PQ893527). This is the first report of avastrovirus infection in psittacine parrots, and is of great veterinary interest. However, the content of this study is a phylogenetic analysis of two viruses, and I think it is best reported as short communication.

【Minor points】

  1. Lines 317-320: Isn't "Wille et al. 2018" a reference (4)? In that case, it should be "Wille et al. 2018 (4)".

Author Response

Lines 317-320: Isn't "Wille et al. 2018" a reference (4)? In that case, it should be "Wille et al. 2018 (4)".

Thank you, this has been corrected.

Round 2

Reviewer 2 Report

Comments and Suggestions for Authors

none

Author Response

Comments 1: None.

Response 1: We thank the reviewer for their effort and time to review our revised manuscript.

Reviewer 3 Report

Comments and Suggestions for Authors

1. Please check line 132 "wefoihjsites with >80% gaps were stripped". wefoihjsites?

2. I recommend that you split the "Next quality" paragraph (line 169) into two: one that (1) describes the assembly of the halachic genome and one that (2) describes the assembly of the parrot genome.
As it stands, it is difficult to understand.

Author Response

Comments 1: Please check line 132 "wefoihjsites with >80% gaps were stripped". wefoihjsites?

Response 1: Sorry, there seems to have been a typing error. This has been corrected to now read "The alignment was trimmed manually and sites with >80% gaps were removed.""

Comments 2: I recommend that you split the "Next quality" paragraph (line 169) into two: one that (1) describes the assembly of the halachic genome and one that (2) describes the assembly of the parrot genome. As it stands, it is difficult to understand.

Response 2: We have added text to the front of each section here to be more explicit which host the assembly refers to. See text: "Firstly, from the galah library, we recovered a genome-length astrovirus-like contig (6,894 bp). Over 50% of the quality filtered, non-host non-rRNA read pairs in the galah library mapped to this sequence, with an abundance of 6.7e04 transcripts per million (TPM). Next, from the king parrot library, we assembled three contigs into a ge-nome-length scaffold (6,825 bp) and finalised the genomic sequence through read mapping."

Reviewer 4 Report

Comments and Suggestions for Authors

It is better organized than the original submitted paper and has no additional comments from my side.

Author Response

Comments 1: It is better organized than the original submitted paper and has no additional comments from my side.

Response 1: We thank the reviewer for their effort and time to review our revised manuscript.